# The Communication from Immune Cells to the Fibroblasts in Keloids: Implications for Immunotherapy

**DOI:** 10.3390/ijms242015475

**Published:** 2023-10-23

**Authors:** Xiya Zhang, Xinfeng Wu, Dongqing Li

**Affiliations:** 1Hospital for Skin Diseases, Institute of Dermatology, Chinese Academy of Medical Sciences and Peking Union Medical College, Nanjing 210042, China; pyszhangxiya@student.pumc.edu.cn; 2Key Laboratory of Basic and Translational Research on Immune-Mediated Skin Diseases, Chinese Academy of Medical Sciences, Nanjing 210042, China; 3Jiangsu Key Laboratory of Molecular Biology for Skin Diseases and STIs, Nanjing 210042, China

**Keywords:** keloid, keloid pathogenesis, keloid microenvironment, immune cells, keloid treatment, immunotherapy, monoclonal antibody

## Abstract

Keloids are a type of fibrotic disease characterized by excessive collagen production and extracellular matrix (ECM) deposition. The symptoms of pain and itching and frequent recurrence after treatment significantly impact the quality of life and mental health of patients. A deeper understanding of the pathogenesis of keloids is crucial for the development of an effective therapeutic approach. Fibroblasts play a central role in the pathogenesis of keloids by producing large amounts of collagen fibers. Recent evidence indicates that keloids exhibit high immune cell infiltration, and these cells secrete cytokines or growth factors to support keloid fibroblast proliferation. This article provides an update on the knowledge regarding the keloid microenvironment based on recent single-cell sequencing literature. Many inflammatory cells gathered in keloid lesions, such as macrophages, mast cells, and T lymphocytes, indicate that keloids may be an inflammatory skin disease. In this review, we focus on the communication from immune cells to the fibroblasts and the potential of immunotherapy for keloids. We hope that this review will trigger interest in investigating keloids as an inflammatory disease, which may open up new avenues for drug development by targeting immune mediators.

## 1. Introduction

Keloids are benign fibroproliferative tumors originating from abnormal wound healing [1]. Many factors can cause keloids, including trauma, surgery, burns, vaccination, acne, and folliculitis, which can be summarized as dermal injury and irritation, in general. However, superficial injuries that do not reach the reticular dermis will not cause keloids, suggesting that keloids result from injury to this skin layer and subsequent abnormal wound healing [2]. Keloids and hypertrophic scars are two types of commonly recognized pathological scars. Keloids have the following three characteristics that distinguish them from hypertrophic scars. First, keloid lesions extend beyond the boundaries of the original lesion and continue to grow from year to year, with little spontaneous regression [1]. Second, fibroblasts from hypertrophic scars and keloids differ. Although hypertrophic scar fibroblasts show a slight increase in basal collagen synthesis, they react normally to growth factors. However, keloidal fibroblasts produce high amounts of collagens, elastin, fibronectin, and proteoglycan and respond abnormally to stimulation. Third, the recurrence rate of keloids is very high [1,3]. Surgical removal of keloids alone has been reported to recur in 70% to 100% of patients and usually results in more intense collagen accumulation and greater lesion formation [4]. Even with surgical resection combined with radiotherapy, the recurrence rate is still high (22%) [5].

The inflammatory, proliferative, and remodeling phases represent three different, although chronologically overlapping, stages of normal wound healing [6]. Keloids are often thought to result from a prolonged proliferative phase and delayed remodeling phase. During normal wound healing, immune cells are recruited during the inflammatory phase to defend against invading microorganisms. Fibroblasts are subsequently stimulated to produce extracellular matrix (ECM) upon which the injured site can be repaired and reshaped [7]. The excess ECM is further digested, and immature collagen III is eventually replaced by mature collagen I [8]. If any steps in this process are disrupted, keloids can develop, which are characterized by abnormal activation of fibroblasts and immune cell infiltration.

Compared with the surrounding tissues, there are more immune cells in the keloid microenvironment, such as macrophages, mast cells, T lymphocytes, and dendritic cells (DCs) [9]. These immune cells were often observed to be in contact with each other or with fibroblasts in paraffin sections or under scanning electron microscopy [9], indicating the close interactions between them. Immune cells release a variety of cytokines and growth factors that promote the deposition of ECM and the proliferation of fibroblasts, which further trigger the progression of keloids [10]. Moreover, a recent study has shown a positive correlation of immune cells with keloid recurrence [11]. This review summarizes the communication from immune cells to the fibroblasts and provides future directions for keloid immunotherapy research.

## 2. Communication from Immune Cells to the Fibroblasts

The microenvironment of keloids is mostly composed of fibroblasts, immune cells, vascular endothelial cells, smooth muscle cells, and the cytokines or growth factors released by them, all of which play an important role in keloid development. The rapid development of single-cell RNA sequencing (scRNA-seq) technology has provided a powerful tool for analyzing the heterogeneity and interaction of fibroblasts and immune cells in the keloid microenvironment (Figure 1).

Normal human dermis fibroblasts can be classified into four subpopulations: secretory–papillary, secretory–reticular, mesenchymal, and pro-inflammatory [12]. Deng et al. [13] found the same to be true for keloidal fibroblasts when using scRNA-seq. Moreover, the proportions of the secretory–papillary, secretory–reticular, and pro-inflammatory subpopulations were diminished, while the proportion of the mesenchymal subpopulation was elevated in keloids [13]. Genes linked with matrix synthesis and fibroblast differentiation, such as ASPN, POSTN, CTHRC1, FN1, and COL11A1, were enriched in the mesenchymal subpopulation [13]. Shim et al. grouped fibroblasts from normal and keloidal tissues into seven subpopulations (FB1 to FB7), including the previously defined subpopulations (secretory–papillary, secretory–reticular, mesenchymal, and pro-inflammatory). The transcriptome profile of keloid fibroblasts (FB1 and FB2) was similar to that of the previously defined mesenchymal fibroblast, and the amount of FB1 and FB2 in keloid tissue was significantly higher than that in normal tissue [14]. Furthermore, Feng et al. also found that the proportion of the mesenchymal subpopulation in keloids was significantly higher than that in surrounding normal skin tissue [15]. Therefore, these studies consistently suggest that mesenchymal fibroblasts are more prevalent in keloids than in normal skin. Thus, the mechanisms for the pathological deposition of ECM in keloids are partially related to the proliferation of mesenchymal fibroblasts, which is possibly driven by fibrogenic growth factors, including transforming growth factor (TGF)-β, fibroblast growth factor (FGF), platelet-derived growth factor (PDGF), vascular endothelial growth factor (VEGF), and POSTN [13,16,17].

Single-cell sequencing analyses also showed that the heterogenicity of immune cells was dramatically changed in keloids. Macrophages were significantly increased in keloids compared to surrounding normal tissue, and two subtypes of macrophages, M1 macrophages and M2 macrophages, both had an effect on keloid formation, but in terms of number, there were more M2 macrophages in keloids than M1 [15]. CD8^+^ T cells were down-regulated in keloid lesions and peripheral blood, in general [18], but two types of CD8^+^ memory T cells, effector memory CD8^+^ T cells and CD103^+^CD8^+^ resident memory T cells, were increased in keloid tissues [19,20]. For CD4^+^ T cells, Rath et al. recorded higher levels of CD4^+^ lymphocytes in keloids [21]. The expression of genes associated with one of its subtypes, Th2 cells, was increased in both keloid lesions and non-lesions. In addition, the CD4^+^:CD8^+^ T-cell ratio was high in keloids when compared to normal skin. Mast cells and dendritic cells, especially inflammatory dendritic cells, were also increased in keloids.

Macrophages. scRNA-seq analysis suggested that there were also substantial differences in the immune profiles between keloids and surrounding tissues, especially the mononuclear phagocytic system. Compared to the surrounding normal tissue, a much higher proportion of macrophages was observed [15]. Under different microenvironments, macrophages could polarize into two major phenotypes: classically activated macrophages (M1) and alternatively activated macrophages (M2) [22]. Feng et al. compared keloid lesions and surrounding normal tissues by using single-cell RNA sequencing and found that the proportion of M2 macrophage subsets in keloid lesions was higher than that of M1 macrophage subsets [15]. The polarization of monocytes into classically activated M1 macrophages was mediated by interferon-γ (IFN-γ), interleukin-12 (IL-12), tumor necrosis factor α (TNFα), damage-associated molecular patterns (DAMPs), and lipopolysaccharide (LPS) [23]. M1 could promote inflammation, eradicate invasive microbes, and promote type I immune responses [24]. M1 macrophages were pro-inflammatory but antifibrotic, as demonstrated by increased pro-inflammatory factor production, such as IL-1β, IL-6, CCL2, CCL7, TNF-α, and increased matrix metalloproteinases (MMPs) to degrade collagen [22,23]. In addition, IL-1β, IL-6, and TNF-α secreted by M1 could stimulate the proliferation of fibroblasts and keratinocytes in addition to their pro-inflammatory effect [22]. Under the stimulation of IL-4, IL-13, or apoptotic neutrophils, alternatively activated M2 macrophages could be transformed from monocytes [22]. M2 macrophages could produce cytokines that stimulate fibroblast proliferation, such as PDGF, VEGF, TGF-β, and insulin-like growth factor (IGF)-1. Interestingly, M1 macrophages predominated in the early phases of scar development (inflammatory and early proliferative phases), while M2 macrophages predominated in the late phases of scar development (late proliferative and remodeling phases) [25]. In fact, it was found that a two-cell circuit symbiosis exists between fibroblasts and macrophages in keloid tissues, which means they interact to obtain mutually required growth factors. By sustaining the expression of CCL2, CSF1, and IL-6, fibroblasts give signals for the chemotaxis, residency, and activation of macrophages in keloids [26]. Macrophages, in turn, produce TGF-β, IGF, VEGF, and PDGF to support the survival and proliferation of fibroblasts [27].

CD8^+^ T cells. Through scRNA-seq, Xu et al. [18] found that the downregulation of CD8^+^ cytotoxic T cells (CTLs) is a feature in the peripheral blood and keloid lesions. There is evidence that CD8^+^ T cells inhibit the proliferation of keloid-associated fibroblasts. Fibroblasts were strongly suppressed by both direct and indirect cultivation of CD8^+^ T cells and fibroblasts, with decreased growth viability and increased apoptosis [28]. Therefore, the abnormal proliferation of fibroblasts in keloids may be related to the decrease in the number of CD8^+^ T cells, which leads to the weakening of the inhibitory effect on keloid fibroblasts, resulting in the accumulation of ECM.

Chen et al. found, by applying flow cytometry, that the numbers of effector memory CD8^+^ T cells and CD103^+^CD8^+^ resident memory T cells were higher in keloid tissues [19,20]. The levels of CD8^+^ tissue-resident memory T cells (CD8^+^ TRM cells) were also discovered to be increased in keloids [20,29]. Although they can protect the body from secondary infection to a certain extent, they also cause continuous inflammatory responses via the overproduction of IFN [20]. Considering the characteristics of a long lifespan and tissue restriction of CD8^+^ TRM cells [30], CD8^+^ TRM cells may be one of the important reasons for keloid recurrence. Even after surgical resection, keloids often form repeatedly because CD8^+^ TRM cells can survive after surgery. Unfortunately, there is currently no way to prevent TRM-mediated keloid recurrence, and most of these cells are invasive [20].

CD4^+^ T cells. Naïve CD4^+^ T cells differentiate into distinct Th lineages, including Th1, Th2, Th17, and Tregs [31]. Wu et al. analyzed gene and protein expression in biopsy specimens obtained from lesional and non-lesional skin of keloid patients and healthy skin. They found that lesional versus normal skin showed significant upregulation of markers of T-cell activation/migration (ICOS, CCR7); the Th2 (IL4R, CCL11, TNFSF4/OX40L), Th1 (CXCL9/CXCL10/CXCL11), and Th17/Th22 (CCL20, S100As) pathways; and JAK/STAT signaling (JAK3). Non-lesional skin also exhibited similar trends [32]. Diaz et al. [33] found upregulated expression of Th2-related genes in lesional and non-lesional skin of keloids and reported that dupilumab targeting the Th2 axis had a significant effect on keloid patients. Th2 cells also mediate pruritus in keloids by producing IL-4 and IL-13 [34].

Boyce et al. revealed that the CD4^+^:CD8^+^ T-cell ratio is high in keloids when compared to normal skin, whereas normal acute wound healing is associated with an initial high CD4^+^:CD8^+^ T-lymphocyte ratio that declines as the lesion heals [35]. Therefore, they inferred that the high CD4^+^:CD8^+^ T-cell ratio may keep the keloid in a high inflammatory response condition as in the early phase of acute wound healing [35]. In the future, treatments that adjust the ratio of CD4^+^:CD8^+^ T cells are expected to reduce keloids, such as breaking the high CD4^+^:CD8^+^ T-cell ratio in keloids.

Mast cells. When applying RNA-seq analysis, Wu et al. suggested that mast cells were increased in keloid lesions compared to non-lesions [32]. Furthermore, Hawash et al. revealed that the number and density of mast cells and their storage granules were increased in pruritic keloids compared to nonpruritic keloids [34]. Mast cells secrete histamine and nerve growth factor (NGF) from preformed granules, forming histamine-driven pruritus [34]. NGF can induce histamine release from mast cells, and histamine enhances further production and secretion of NGF by keratinocytes, which results in a cyclic relationship between histamine and NGF, forming histamine-driven pruritus [34]. Fibroblasts also induce the release of histamine [36,37]. Therefore, the above evidence supports the use of antihistamines or NGF inhibitors to relieve pruritus in keloid patients. In addition, mast cell degranulation also releases VEGF; TGF-β; TNF-α; the pro-inflammatory cytokines IL-1, IL-6, and IL-33; and the chemokine chymase [10,38]. TNF-α, IL-1, and IL-33 play a significant part in the start of the host defensive cascade that results in fibrosis [37]. TGF-β can stimulate fibroblasts, and then it promotes the process of ECM deposition. Chymase boosts keloid fibroblast proliferation and collagen production by activating TGF-β and triggering SMAD signaling [1,39]. Thus, mast cells exert a significant effect on keloid fibroblasts through the action of various cytokines.

DCs. New evidence suggests that DCs can impact the function of fibroblasts and the vasculature [40]. Wu et al. discovered an increase in dendritic cells (CD11c+) in keloids, including dendritic cell infiltrates that are typical of AD [32]. Rath et al. [21] found that monocytic-derived inflammatory DCs are the main infiltrating DCs in keloid tissue [21]. The correlation between inflammatory DCs and fibroblasts was higher in keloid tissue than in surrounding or healthy tissue. During wound healing, keloids could recruit and trigger inflammatory DC precursors, that is, monocytes, causing inflammation without presenting autoimmune antigens [21]. Inflammatory DCs may stimulate the expression of metalloproteinases in fibroblasts, including ADAM10, ADAM17, and CD10. ADAM10 can promote fibroblast activation and fibrosis through sEphrin-B2, which was identified as a novel profibrotic mediator of lung and skin fibrosis [41]. ADAM17 has been implicated in inflammation [21]. Most immunological effectors, including cytokines and chemokines, are expressed as inactive precursors and need to be processed by an activated protease. In this context, ADAM10 and 17 are of particular importance for cellular differentiation and proliferation, cleaving important mediators such as proTNF and the Notch receptors, leading to fibroblast proliferation and the production of ECM components [21]. Therefore, these proteases may be effective therapeutic targets. Protease inhibitors have been developed, and clinical trials are underway and have been marketed [42].

Immune cells invade and regulate the microenvironment by secreting cytokines [19], which play a key role in the keloid microenvironment [43]. Numerous cytokines, such as IL-4, IL-6, IL-10, IL-12, IL-13, IL-17, TNF-α, and TGF-β, are markedly elevated in the keloid local microenvironment [43,44]. They are engaged in the prolonged proliferative phase of keloids in addition to their roles in the inflammatory phase [45]. In contrast, serum concentrations of IFN-α, IFN-γ, and TNF-β, which are molecules that suppress collagen synthesis and fibroblast proliferation, are lowered, resulting in a loss of inhibition and consequent encouragement of uncontrolled collagen production [1,46]. Targeting these key cytokines is expected to improve the keloid treatment dilemma.

## 3. Immunotherapy of Keloids

Current treatments for keloids include surgical resection, radiotherapy, injections, cryotherapy, laser, radiofrequency ablation, and various drug therapies. However, none of the drugs are keloid-specific [47]. In addition, few current treatments can completely prevent recurrence. The recurrence rate is 70% to 100% for surgical resection alone [4] and 22% for surgical resection plus radiotherapy [5]. Therefore, more effective treatment methods need to be developed. Above, we summarized the pathogenic role of various immune cells in keloids, indicating that immunotherapy may be a breakthrough to solve the problems associated with keloid treatment. Here, we summarize the potential targets for immunotherapy in keloids (Figure 1 and Table 1).

### 3.1. TGF-β

As the central cellular effector of fibrotic responses, TGF-β is produced and secreted by inflammatory cells, particularly macrophages, as well as fibroblasts and platelets [1]. Types 1, 2, and 3 are the three subtypes of TGF-β. The levels of type 3 TGF-β are not increased in keloids, but those of types 1 and 2, which activate fibroblasts and are involved in fibrosis and inflammation, are increased [57]. In keloids, TGFβ-1 has been associated with increased collagen and fibronectin synthesis by fibroblasts. The addition of TGFβ-1 to keloidal fibroblasts leads to increased synthesis of procollagen RNA [46]. TGF-β is involved in both suppressive and inflammatory immune responses. Under steady-state conditions, TGF-β regulates thymic T-cell selection and maintains homeostasis of the naïve T-cell pool. TGF-β inhibits cytotoxic T-lymphocyte (CTL), Th1, and Th2 cell differentiation while promoting peripheral (p)Treg, Th17, Th9, and Tfh cell generation and T-cell tissue residence in response to immune challenges. Similarly, TGF-β controls the proliferation, survival, activation, and differentiation of B cells, as well as the development and functions of innate cells, including natural killer (NK) cells, macrophages, dendritic cells, and granulocytes [58]. TGF-β promotes keloid formation by regulating fibroblasts and macrophages. TGF-β works in tandem with IL17A to stimulate IL-6 and CCL2 production in fibroblasts, which are macrophage chemokines and offer a way for macrophages to be recruited to the keloid microenvironment [27]. Considering macrophages are an important source of TGF-β production, the recruited macrophages produce more TGF-β and further promote collagen synthesis in the keloid microenvironment. In addition, by co-culturing fibroblasts with Tregs, Chen et al. [8] found that Tregs promoted collagen expression, which was more pronounced in keloids than in non-keloid controls, and that the process required TGF-β and anti-CD3/CD28 stimulation. Therefore, targeting TGF-β-related receptors or pathways is a promising method for the treatment of keloids. Current therapies targeting TGF-β/Smad signaling in fibroblasts can be divided into genetic, cellular, and pharmacological therapies [59]. Several investigational agents that target the TGF-β pathway have entered clinical trials. Clinical trials of the TGF-β neutralizing antibody fresolimumab (GC1008) are underway [48].

Fresolimumab is a pan-TGF neutralizing antibody that targets all three TGF-β isoforms and is being tested in clinical trials for a variety of fibrotic and cancer disorders [48]. Phase 1 clinical studies have been conducted in patients with focal segmental glomerulosclerosis (FSGS), idiopathic pulmonary fibrosis (IPF), advanced malignant melanoma, and renal cell carcinoma (RCC). The administration of fresolimumab was well tolerated in three separate human phase 1 clinical trials, and no treatment-related serious adverse effects were reported [60]. In fibrotic diseases, Rice et al. found that fresolimumab treatment reduced biomarkers and improved clinical symptoms in patients with systemic sclerosis (SSc) [49]. The expression of several TGF-β and collagen-related genes, including CTGF, SERPINE1, and COL10A1, declined after fresolimumab treatment compared with baseline [49]. In addition, fresolimumab treatment was associated with a rapid, dramatic decline in fibroblast infiltration of the deep dermis [49]. Since fibroblasts are the main pathogenic cells of keloids, fresolimumab may be a novel therapeutic strategy for keloids.

However, persistent inhibition of TGF-β function may lead to side effects because TGF-β is involved in a variety of biological processes [48]. It has been shown that systemic reduction in TGF-β activity can be antifibrotic but also blocks its anti-inflammatory function, leading to exacerbation of inflammation [61]. Adverse effects also include bleeding episodes, such as gastrointestinal bleeding, gingival bleeding, or epistaxis [49]. Local injection of TGF-β antibodies or inhibitors may be an alternative treatment for keloids. Fibronectin extra domain A (Fn-EDA) is a component of the ECM and is specifically expressed in the fibrotic region. McGaraughty et al. designed a dual-specific antibody targeting Fn-EDA to deliver TGF-β antibody to fibrotic kidney lesion sites [50]. The molecule partially targets Fn-EDA and partially neutralizes TGF-β using dual variable domain Ig (DVD-Ig) technology. Systemic injection of bispecific antibodies led to notably higher levels of each molecule in obstructed kidneys than in non-obstructed kidneys, ipsilateral kidneys of sham animals, and other tissues. Since Fn-EDA is also enriched in keloids [1], the bispecific molecule targeting Fn-EDA and TGF-β can also be tried in the treatment of keloids. Bispecific antibody treatment may be more effective and have fewer side effects than the anti-TGF-β antibody alone.

### 3.2. NKG2A/CD94

CD8^+^ T cells can significantly inhibit fibroblast activity and proliferation [28]. Through scRNA-seq, Xu et al. [18] found that the downregulation of CD8^+^ CTLs is a feature in the peripheral blood and keloid lesions. In addition, scRNA-seq also showed specific upregulation of the NKG2A/CD94 complex, which may contribute to the reduction in CTLs within the tissue boundaries of keloid lesions. Therefore, it can be inferred that NKG2A/CD94 indirectly promotes the activity and proliferation of fibroblasts in keloids, thereby exacerbating keloid progression.

Monalizumab is a novel checkpoint inhibitor targeting the NKG2A/CD94 complex in tumor immunotherapy [51]. Monalizumab can enhance NK cell activity against various tumor cells and rescue CD8^+^ T-cell function in combination with programmed-cell-death axis (PDx) blockade [51]. Monalizumab enhances tumor immunity by blocking inhibitory NKG2A receptors to promote CD8^+^ T-cell effector function [51]. Given that CD8^+^ T cells inhibit fibroblast proliferation and activity, monalizumab is a promising agent for reducing excessive ECM in keloids by restoring the inhibitory effect of CD8^+^ T cells on fibroblasts. Furthermore, a prominent advantage of monalizumab is that blocking NKG2A has little toxicity and, in particular, produces no signs of autoimmunity [51]. The most common adverse events of monalizumab were fatigue (17%), pyrexia (13%), and headache (10%). Other rare adverse events were interstitial lung disease, colitis, and hypophosphatemia [51]. In the future, clinical trials of monalizumab in the treatment of keloids may lead to surprising discoveries.

### 3.3. IL-4/IL-13 Pathway

Th2 cells and their cytokines play an important role in the pathogenesis of keloids. Wu et al. [32] found that markers of Th2, including IL4R, CCR5, CCL11, and TNFSF4/OX40L, were significantly upregulated in both keloid lesional and non-lesional skin compared to normal skin. IL-4 and IL-13 not only promote ECM deposition, inflammation, and pruritus of keloids but also induce the transformation of monocytes into M2 macrophages [62], thus playing a pivotal role in the pathogenesis of keloids. IL-4 and IL-13 can also directly stimulate nerve fibers via IL-4 receptors to induce pruritus. Moreover, TGF-β induces the production of IL-31, a Th2-related cytokine that is elevated in skin wound tissue and is significantly correlated with pruritus intensity [34]. This evidence suggests that Th2 cells and their cytokines, especially IL-4 and IL-13, contribute to keloid development, and the Th2 axis may become a breakout area of research in keloid pruritus treatments.

Dupilumab, which was approved for the treatment of moderate-to-severe atopic dermatitis (AD), inhibits type 2 inflammation via the IL-4/IL-13 pathway, also known as the Th2 axis [63]. Interestingly, AD is an independent risk factor for the development of keloids [64]. The higher risk of keloid formation in AD patients suggests an association between keloids and the Th2 axis. Diaz et al. [33] reported a new use of dupilumab in the treatment of keloids. In a patient with both severe AD and keloids, after 7 months of treatment with a 300 mg dupilumab subcutaneous injection every 2 weeks, AD improved significantly, with a 50% reduction in the total size of the fibrotic plaque, shrinking of the large keloid, flattening of the surrounding margins, and full removal of the smaller keloid. Furthermore, dupilumab relieved pruritus in a phase 3 study of patients with moderate-to-severe AD, indicating that it plays a role in blocking signals from itch neurons [52] and may also improve pruritus of keloids. Injection site reactions are the most common adverse event associated with dupilumab. Other relatively rare adverse events include ocular complications (e.g., dry eyes, conjunctivitis, blepharitis, keratitis, and pruritus), head and neck dermatitis, onset of psoriatic lesions, progression of cutaneous T-cell lymphoma, alopecia areata, hypereosinophilia, and arthritis. Most of these adverse effects can be controlled during continued treatment with dupilumab, but some (such as severe conjunctivitis) may result in discontinuation of treatment [53,54]. Above all, dupilumab, which targets the IL-4/IL-13 pathway, may provide a better treatment choice for keloid patients, especially those with concurrent AD, and is worthy of further large-scale clinical trials.

### 3.4. TSLP

Thymic stromal lymphopoietin (TSLP) is an epithelial-cell-derived cytokine that directly stimulates Th2 cytokine production, leading to Th2 inflammation [65], and collagen production, leading to skin fibrosis [55,66,67]. Studies have shown that IL-13 induces skin fibrosis in AD via TSLP [66]. In IL-13 transgenic mice, TSLP neutralization or genetic deletion of TSLPR led to a considerable decrease in fibrocytes and cutaneous fibrosis [66]. Given that TSLP expression is also greatly enhanced in keloid lesions, extending to uninvolved skin [32], it is meaningful to investigate whether TSLP also has a role in promoting fibrosis in keloids. Tezepelumab is a human monoclonal antibody that blocks TSLP, which is also implicated in the pathogenesis of asthma by driving Th2 inflammation of the airway. Clinically, tezepelumab is used to treat severe, uncontrolled asthma in adults and adolescents [55]. The most common adverse events are nasopharyngitis, upper respiratory tract infection, headache, and asthma (which was more frequently observed in the placebo group than in the tezepelumab group) [55]. Tezepelumab can decrease the enhanced immune cells in keloid tissues. Although there have been no clinical trials of tezepelumab in keloids, it is worth assessing its effectiveness as a treatment of keloids that may produce similar effects to dupilumab.

### 3.5. IL-6

IL-6, secreted by M1 macrophages and mast cells [10,68], promotes chemotaxis, residency, and activation of macrophages [26]. Under stimulation of IL-6 and adenosine receptors, macrophages can generate a large amount of IL-10, TGF-β, and VEGF [26]. IL-6 has been shown to induce collagen and α-smooth muscle actin (α-SMA) expression in dermal fibroblasts [56]. Previous research has shown that IL-6 expression is increased in keloid fibroblasts [69]. Inhibiting IL-6 or IL-6 receptor α (IL-6Rα) in keloid fibroblasts revealed a dose-dependent decrease in collagen type I α 2 and fibronectin 1 mRNAs [70]. Furthermore, the mRNA and protein expression levels of gp130 and several downstream targets in IL-6 signaling (JAK1, STAT3, RAF1, and ELK1) were upregulated in keloid fibroblasts versus normal fibroblasts [70]. These findings suggest that IL-6 signaling may play an important role in keloid pathogenesis and provide clues for strategies targeting IL-6 for keloid therapy and prevention. Immunotherapies that target IL-6 signaling, such as tocilizumab, sarilumab, and siltuximab [56], should be further investigated in keloids.

Tocilizumab blocks the IL-6 receptor (IL-6R) and is approved in more than 100 countries worldwide for the treatment of rheumatoid arthritis (RA) and juvenile idiopathic arthritis (JIA) [56]. In fibrotic diseases, the 2018 ACR conference reported the results of a phase III randomized controlled trial of tocilizumab in systemic sclerosis (SSc) treatment [56], in which 210 patients with SSc received 162 mg subcutaneous tocilizumab treatment. Although the primary endpoint was not met at the end of the experiment, the forced vital capacity of patients taking tocilizumab was superior to that of patients taking a placebo, indicating that tocilizumab significantly inhibited the worsening of respiratory disorders due to SSc-induced pulmonary fibrosis. On this basis, tocilizumab is also a valuable candidate for skin fibrosis treatment. In terms of adverse events, the most common adverse events associated with long-term use (96 weeks) were infections, such as pneumonia, infectious tenosynovitis, sepsis, and otitis media [71]. A decrease in neutrophils is often observed during treatment with tocilizumab in RA patients; however, neutropenia is not due to the myelotoxic effects of tocilizumab but rather to changes in the localization of circulating neutrophils [56].

## 4. Future Directions

In this paper, we reviewed the immune microenvironment composed of a variety of immune cells and cytokines involved in the pathogenesis of keloids, including M1 and M2 macrophages, CD8^+^ cytotoxic T cells, Tregs, TRM cells, mast cells, and DCs. In addition, many cytokines, such as TGF-β, IL-3, IL-6, and IL-14, are also important factors in promoting keloid formation. Since the current treatment methods for keloids have difficulty in preventing recurrence, and since the characteristics of keloids as a systemic disease have been increasingly confirmed, exploration of the systemic application of immunotherapy for keloids has become a new treatment breakthrough [59,72,73]. Therefore, this article summarized the important pathogenic pathways or essential cytokines in keloids and reviewed the prospects of targeting them in the treatment of keloids. However, comprehensive studies are needed to compare immunotherapy with other traditional methods in terms of efficacy, safety, cost, and other relevant metrics. In terms of effectiveness, as no immunotherapy drugs described in this paper have yet been applied in clinical trials on keloids, there are no specific data on therapeutic effectiveness at present, and relevant clinical trials need to be conducted in the future to fill this gap. In terms of safety, the adverse events of each immunotherapy drug mentioned in this review have been described separately in the corresponding sections. Compared with existing treatments such as surgical resection, cryotherapy, or laser, immunotherapy is mainly administered orally or by injection, which causes less skin damage, improves the patient’s treatment experience, and reduces the risk of infection, bleeding, and recurrence at the lesion site. In terms of cost, since most immunotherapy drugs have not yet been developed for keloid indications, there is no clear reference for the price, but due to the high cost of immunotherapy drug research and development, it is estimated that the price of immunotherapy may be higher than the current traditional treatment.

There are still many deficiencies in the recent research on keloids. The understanding of the immune aspect of keloids is neither specific nor comprehensive. First, many specific mechanisms are still vague and need to be clarified. For example, many immune cells and immunotherapies have been well studied in cancer but not in keloids. Our understanding of the role of these cells in keloids is partly derived from their function in cancer, which is not sufficiently rigorous. Second, mechanisms of some important features in keloids, such as high recurrence and pruritus, are incomplete and need to be supplemented. Moreover, many studies on keloids are limited due to the lack of reliable and reproducible keloid models. At present, there are no large clinical trials of monoclonal antibodies applied in the treatment of keloids, resulting in a gap in the field of keloid immunotherapy.

Future research on keloid immunity will be informed by the methods of tumor immunity research, but it is necessary to identify the specific mechanism of keloids. Take CD8^+^ T cells, for example, which can kill tumor cells and inhibit keloid fibroblast proliferation; studies have shown that clonal expansion of the effector and exhausted tumor-infiltrating CD8^+^ T cells is observed for various human cancers [74], while the number of CD8^+^ T cells in the keloid microenvironment and peripheral blood is reduced compared to in healthy people [18].

Keloids are prone to relapse, and the current findings conclude that multiple cells and factors are involved, including CD8^+^ TRM cells, Tregs, and sHLA-E. Further studies are needed to characterize the role(s) of different cell types and the epigenetic factors in the recurrence of keloids. 

Currently, there is no animal model that can fully capture the features of keloids in humans. Perhaps an alternate way is to try using a pressure load device in different species to mimic the tension in keloids [75,76]. Another approach is to transplant human keloid xenografts into immunodeficient mouse strains in which the keloid tissue can survive for several weeks [77,78].

Clinical studies of multiple immunotherapies should be carried out in the future. A variety of studies have shown that single therapy is not as good as combination therapy for keloids. For example, when intralesional steroids are combined with surgical excision, the recurrence rate appears to be significantly lower than that with surgery alone [46]. In addition, radiation therapy is often used as a combination treatment option. Radiation therapy is usually administered 24–48 h after surgical resection as an adjunct, and the recommended dose of radiation is 40 Gray, divided into several treatments to minimize adverse effects. The mechanism of radiation therapy for keloids is anti-angiogenesis and continuous anti-fibroblast activity. Inhibition of angiogenesis reduces the delivery of inflammatory cytokines, and continuous inhibition of fibroblast activity leads to reduced collagen synthesis, thereby inhibiting the development of keloids [79]. Therefore, the combination of systemic immunotherapy with other treatment methods, such as surgical resection, radiation therapy, or intralesional injection, is also a new promising future direction.

## 5. Conclusions

A keloid is the result of the joint action of fibroblasts and their surrounding immune cells. Immune cells penetrate the microenvironment and secrete cytokines to stimulate fibroblasts. We focused on the communication from immune cells to the fibroblasts and the potential of keloid immunotherapy. Based on published single-cell sequencing data, we summarized the characteristics of different immune cells in keloids and their effects on fibroblasts. These immune cells exert crucial roles in the pathogenesis of keloids by secreting many cytokines or growth factors to stimulate the fibroblasts. Future studies are anticipated to compare the effects of different immunotherapies on keloids.

## Figures and Tables

**Figure 1 ijms-24-15475-f001:**
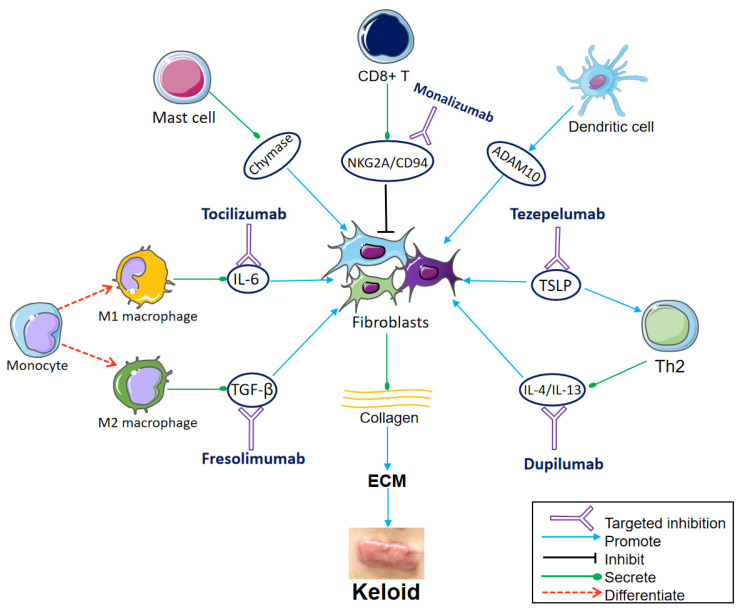
Communication from immune cells to the fibroblasts and important targets for immunotherapy. Fibroblasts differentiate into different subsets, producing large amounts of collagen, which aggregates to form ECM and eventually keloids. The monocytes differentiate into M1 macrophages and M2 macrophages under different conditions. M1 secretes interleukin (IL)-6, which induces collagen expression in dermal fibroblasts. Tocilizumab blocks the IL-6 receptor. M2 secretes transforming growth factor (TGF)-β, which promotes the differentiation and proliferation of fibroblasts. Fresolimumab targets TGF-β, decreasing the expression of collagen-related genes and fibroblasts infiltration. Mast cell degranulation releases chemokine chymase, which boosts keloid fibroblast proliferation. CD8^+^ T cells inhibit fibroblasts. The NKG2A/CD94 complex is expressed on the surface of CD8^+^ T cells and suppresses CD8^+^ T cells. Monalizumab is expected to restore the inhibitory effect of CD8^+^ T cells on fibroblasts by blocking NKG2A/CD94. Dendritic cells stimulate the expression of ADAM10 in fibroblasts. Thymic stromal lymphopoietin (TSLP) directly stimulates Th2 cytokine production, leading to Th2 inflammation and collagen production. The TSLP antibody tezepelumab is also worth trying in keloid treatment. IL-4 and IL-13, produced by Th2 cells, promote ECM deposition, inflammation, and pruritus of keloids. Dupilumab inhibits type 2 inflammation by targeting IL-4/IL-13. (Parts of the figure were drawn by using pictures from Servier Medical Art, which is licensed under a Creative Commons Attribution 3.0 Unported License (https://creativecommons.org/licenses/by/3.0/ (accessed on 10 October 2023)).

**Table 1 ijms-24-15475-t001:** Summary of potential immunotherapies for keloids.

Name of Drug	Target	Immunomodulatory Mechanism	Effects on Fibrosis	Common Side-Effects	Ref.
Fresolimumab	TGF-β	---	Inhibits skin fibrosis of SSc.	Promotes inflammation; Bleeding	[48,49]
Bispecific antibody	Fn-EDA and TGF-β	---	Inhibits renal fibrosis.	---	[50]
Monalizumab	NKG2A/CD94	Restores the inhibitory effect of CD8^+^ T cells on fibroblasts.	---	Fatigue; Pyrexia; Headache	[51]
Dupilumab	IL-4/IL-13 pathway	Inhibits Th2 immune inflammation.	Reduces keloid plaque.	Injection site reactions	[33,52,53,54]
Tezepelumab	TSLP	Inhibits Th2 immune inflammation. Decreases the enhanced immune cells in keloid tissues.	---	Nasopharyngitis; Upper respiratory tract infection; Headache; Asthma	[55]
Tocilizumab	IL-6 receptor	---	Inhibits SSc-induced pulmonary fibrosis.	Infection; Neutropenia	[56]

TGF-β: transforming growth factor β; Fn-EDA: fibronectin extra domain A; AD: atopic dermatitis; TSLP: thymic stromal lymphopoietin; SSc: systemic sclerosis; ECM: extracellular matrix.

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
