# Peer review of "The Communication from Immune Cells to the Fibroblasts in Keloids: Implications for Immunotherapy"

_ijms, 2023, doi:10.3390/ijms242015475_

Round 1

Reviewer 1 Report

This review entitled " The crosstalk between fibroblasts and immune cells in keloids: Implications for immunotherapy " descripted briefly keloid formation focusing on the crosstalk between fibroblasts and immune cells, moreover, the involvement of various cytokines was included; which made the understanding that keloids are the outcome actions of fibroblasts and surrounding immune cells in joint. In other words, they are due to imbalances of cells’ interactions in the microenvironment leading keloids formation.

In general, this article provides new perspective of keloids’ pathogenesis and clinical therapeutic implication, it is a valuable reference for investigating keloids. However, in my point of view, there are some impertinent words in text should be revised, such as “keloidal fibroblasts generate large quantities of collagen…. (Line 37)” the “quantities” used is not suitable, ….and so on.

Reviewer 2 Report

The work gives a good idea of state of art about keloids and the possible immunological mechanisms involved. It also shows  " some but not all" possible immunotherapeutic  approaches and the target under study. The work is well organized and comprehensively described. The wark is scientifically sound and the references are appropiated. Good inglesh

Reviewer 3 Report

With this review article, the authors provide a very current, information-rich perspective on keloids. It is thought that with modifications, this would be a helpful addition to the literature about the presence and roles of immune cells in keloids, which also highlights potential novel therapeutic targets.

One general concern is about vague/superficial content about fibroblasts (e.g. diversity not mentioned in abstract or pg2; what is an “activated” fibroblast; narrow focus on myofibroblasts, whose relevance in keloid is controversial, is over-emphasised whilst other subsets are neglected). The strength of this review is in the immune cell content and their influence on fibroblasts. It is suggested to remove the “cross talk” between the cell types and focus solely on the one-way communication from immune cells TO the fibroblasts. This feels consistent with the “immunotherapy” angle, which by definition is a therapy that modulates the immune system/cells. This adjustment would affect the content, as well as the figure (specific suggestions at the bottom of this review).

(Somewhat chronologically) Fibroblast comments (pg2-3)

P3-103 – related to the _expansion_ of mesenchymal… which has been linked to … (not predominantly).

Line 107 – remove “myo” from fibroblast (this devalues the heterogeneity highlighted by these single-cell studies)

Lines 106-113 – these lines are unsubstantiated and the concluding statements don’t follow.

(As mentioned, it is suggested to minimise the fibroblast section and focus on immune influence on fibroblasts (potentially all subtypes, not just “myo”fibroblasts). Clarify “precursor effector” (line 180).

The immune cell section (~line 125 onwards), although the ratio of M1/M2 is mentioned, how do the absolute numbers of each compare to controls? And what are the controls for these studies (i.e. are these findings relevant to all scars or specifically keloids?). Line 127 – M1 are also crucial to scarring fibrosis (e.g. Lucas et al 2010 and others).

In the immunotherapy section – reconsider the title of Table 1. Add a column to table 1 for effects on fibrosis (if they exist) but in mechanism column, only include the immunomodulatory mechanisms (potentially this will mean the removal of TGFb, Fn, FSTL1). A separate side-effect column could be made if that is of interest here, rather than just in the body of the text. 

If only focusing on immunomodulatory immunotherapies, this may influence the placement (and quantity) of info on TGFb in the body of the text (200-250). Consider grouping all Th2 related content. The IL6 content should be adjusted to highlight it is a product of xyz immune cell(s), and the efficiency of the current content considered. Same comment for FSTL1 – currently this section doesn’t even mention immune cells; just because it is an antibody therapy, does not make it an “immunotherapy” (until you tie it to immunomodulation as the mechanism of action). Is there nothing to mention here about targeting macrophages (of any variety)? This feels lacking given its emphasis on pg3-4.

Finally, we suggest that the organisation may benefit from reconsideration, as there are places where the content is somewhat disjointed. One potential approach would be to shuffle the order to first summarise what scRNA datasets have shown – thoroughly, acknowledging variations between studies. For example, increased fibroblasts overall (then what subsets), and whether or not each paper equally observed increased immune cell infiltration (a few examples of discrepancies are mentioned below). Once the relative abundances of various cell populations are presented, the narrative could then progress into which immune cells are increased and what they’re expressing and how these can act on fibroblasts/the environment to promote keloid pathology.

Examples of discrepancies: Lines 137-139 state an increased CD8+ population, with high CD4:CD8 ratio, but this is followed by a statement that CD8s suppress fibroblasts, and another paper finds a decrease in CD8s in keloids. The text next discusses CD4-Th2 lineages, but earlier text has somewhat discounted them due to low numbers. Please report these variations and address them “head on”.

Throughout – check referencing, avoid repetition. 

Figure suggestions – it is very unusual to have such an enormous amount of new text within the figure legend. The content should be moved to the main body of the text, and cited properly (and probably its placement further along the paper is appropriate). Other suggestions for the figure (consistent with the comments above) include: create a single “scar fibroblast” or a few different “scar fibroblasts” (perhaps of different colours to represent their potentially different roles) as the receiver of immune cell signals. 

Group TSLP with Th2

Reconsider the display of the monocyte-macrophages – M1 off to the left looks unimportant, TRM with M2 phenotype not captured.

Remove the TGFb content to the right – unless it appears as an immune cell product (otherwise it isn’t part of the focus of the paper). Same comment for FSTL1 – focus on its production from immune cells. 

Which (immune) cells produce the IL6? Remove any arrows pointing away from the fibroblasts (this is too incompletely covered and doesn’t need to be the focus of this paper).

Fine.

Reviewer 4 Report

ijms-2606789, The crosstalk between fibroblasts and immune cells in keloids: 2 Implications for immunotherapy by Xiya Zhang et al.

Comments:

Abstract: 

·       The abstract starts with "Keloids are fibrotic diseases." While technically accurate, it might be clearer to say, "Keloids are a type of fibrotic disease."

·       Instead of "seriously affect patients’ quality of life," consider rephrasing to "significantly impact the quality of life of patients."

·       The concluding sentence could be strengthened by providing more specifics. For example, suggesting particular avenues of drug development or highlighting any recent breakthroughs in understanding keloids as an inflammatory disease.

1. Introduction:

·       The term "ECM" should be fully expanded upon its initial use for clarity.

2. Crosstalk between fibroblasts and immune cells

·       Ensure all abbreviations are introduced before being used. For instance, EDA , TGF-β, FSTL-1,..

·       The factors leading to the polarization of M1 and M2 macrophages in the keloid environment could be expanded upon.

·       The discussion about the balance between CD4+ and CD8+ T cells in keloids vs. normal skin is insightful. Still, the implications of this balance for therapeutic strategies could be elaborated on.

·       The role of mast cells in pruritic keloids is an interesting inclusion. Since pruritus can be a significant symptom for patients, a deeper dive into potential therapeutic strategies targeting this aspect could be beneficial.

3. Immunotherapy of keloids

·       The reviewer suggests providing a brief introductory sentence for each drug, which gives an overview before delving into the specifics.

·       While some drugs like “Fresolimumab” have their adverse effects discussed, ensure a similar approach for other drugs. Understanding potential side effects is crucial when considering therapeutic viability.

·       The reviewer suggests highlighting how immunotherapy for keloids compares to other treatments in terms of efficacy, safety, cost, and other relevant metrics. 

4. Future directions:

·       The section seems to jump between several ideas without a clear transition. It would help to organize the challenges, future directions, and potential solutions into separate subheadings.

·       The reviewer suggests highlighting any contrasting findings between keloid and cancer research to emphasize the unique challenges associated with keloids.

·       It would be valuable if the authors could propose or speculate on potential solutions, even if preliminary, based on the recent findings or parallel research areas.

·       The mention of the potential benefits of combining immunotherapy with other treatments like surgical resection is intriguing. It would be helpful to elaborate more on how these combined therapies might work together, potentially citing any early research or preliminary findings.

5. Conclusion:

·       The conclusion would benefit from a more detailed summary that encapsulates the primary discoveries or significant insights discussed throughout the manuscript. Highlighting these central points will provide readers with a clearer understanding of the article's core contributions."

·       The use of the phrase "It is hoped" can be rephrased for a more assertive tone, e.g., "Future studies are anticipated to reveal..."

Moderate editing of English language required

Round 2

Reviewer 4 Report

I would like to thank the authors for carefully addressing all of the reviewer's comments.